# Hydrogen Peroxide Promotes the Production of Radiation-Derived EVs Containing Mitochondrial Proteins

**DOI:** 10.3390/antiox11112119

**Published:** 2022-10-27

**Authors:** Caitlin E. Miller, Fangfang Xu, Yanming Zhao, Wei Luo, Weixiong Zhong, Kristy Meyer, Rani Jayswal, Heidi L. Weiss, William H. St. Clair, Daret K. St. Clair, Luksana Chaiswing

**Affiliations:** 1Department of Toxicology and Cancer Biology, University of Kentucky, Lexington, KY 40506, USA; 2Department of Radiation Medicine, University of Kentucky, Lexington, KY 40506, USA; 3Department of Pathology and Laboratory Medicine, University of Wisconsin-Madison, Madison, WI 53706, USA; 4Biostatistics Shared Resource Facility, Markey Cancer Center, University of Kentucky, Lexington, KY 40506, USA

**Keywords:** hydrogen peroxide, radiation resistance, mitochondria, extracellular vesicles

## Abstract

In spite of extensive successes, cancer recurrence after radiation treatment (RT) remains one of the significant challenges in the cure of localized prostate cancer (PCa). This study focuses on elucidating a novel adaptive response to RT that could contribute to cancer recurrence. Here, we used PC3 cell line, an adenocarcinoma from a bone metastasis and radio-resistant clone 695 cell line, which survived after total radiation dose of 66 Gy (2 Gy × 33) and subsequently regrew in nude mice after exposure to fractionated radiation at 10 Gy (2 Gy × 5). Clone 695 cells not only showed an increase in surviving fraction post-radiation but also an increase in hydrogen peroxide (H_2_O_2_) production when compared to PC3 cells. At the single cell level, confocal microscope images coupled with IMARIS rendering software demonstrate an increase in mitochondrial mass and membrane potential in clone 695 cells. Utilizing the Seahorse XF96 instrument to investigate mitochondrial respiration, clone 695 cells demonstrated a higher basal Oxygen Consumption Rate (OCR), ATP-linked OCR, and proton leak compared to PC3 cells. The elevation of mitochondrial function in clone 695 cells is accompanied by an increase in mitochondrial H_2_O_2_ production. These data suggest that H_2_O_2_ could reprogram PCa’s mitochondrial homeostasis, which allows the cancer to survive and regrow after RT. Upon exposure to RT, in addition to ROS production, we found that RT induces the release of extracellular vesicles (EVs) from PC3 cells (*p* < 0.05). Importantly, adding H_2_O_2_ to PC3 cells promotes EVs production in a dose-dependent manner and pre-treatment with polyethylene glycol-Catalase mitigates H_2_O_2_-mediated EV production. Both RT-derived EVs and H_2_O_2_-derived EVs carried higher levels of mitochondrial antioxidant proteins including, Peroxiredoxin 3, Glutathione Peroxidase 4 as well as mitochondrial-associated oxidative phosphorylation proteins. Significantly, adding isolated functional mitochondria 24 h prior to RT shows a significant increase in surviving fractions of PC3 cells (*p* < 0.05). Together, our findings reveal that H_2_O_2_ promotes the production of EVs carrying mitochondrial proteins and that functional mitochondria enhance cancer survival after RT.

## 1. Introduction

Prostate cancer (PCa) is one of the most diagnosed cancers in men besides skin cancer [1]. This cancer arises from the overgrowth of the epithelial lining within the prostate [2]. It is estimated that there will be around 268,490 cases diagnosed in the US in 2022 [3]. Not only is this cancer one of the most diagnosed in men, but it is the second leading cause of cancer death in men. It is estimated that 11% of male cancer deaths will be attributed to PCa. Both of these percentages have steadily increased over recent years [1,4,5]. Viable treatment options for prostate cancer include surgery, radiation, and hormone therapy [6].

Radiation is a widely used treatment for many cancers. Around 50% of PCa patients will receive radiation over the course of their treatment [7]. Radiation kills cancer cells via at least two mechanisms; (1) The direct effect, by directly causing ionization of macromolecules and (2) indirect effect, by producing reactive oxygen species (ROS) such as the superoxide anion, hydroxy anion, and H_2_O_2_ through radiolysis of water [8,9]. ROS production specifically within the mitochondria can further stimulate ROS production by increasing mitochondria membrane permeability [9]. The ROS produced by the indirect effect can also cause DNA breaks, further damaging the cell through increasing mitochondrial membrane permeability, causing protein misfolding, and lipid peroxidation [9,10]. Due to tumor heterogeneity, it is difficult for radiation to target every cell within these tumors. Although a widely used treatment, some cases of PCa will recure after radiation. For the purpose of our study, this will be called radiation resistance. The mitochondrial-mediated mechanism of how these cancers become resistant is debated. Some argue that alterations in mitochondria can lead to radiation sensitivity, while others conclude that mitochondria can contribute to radiation resistance [11,12].

H_2_O_2_ is one of the ROS that is produced during radiation treatment (RT) [9]. H_2_O_2_ possesses a longer half-life (2.2 s versus 10^−6^ and 10^−9^ s) compared to the other ROS produced during radiation treatment [13,14,15,16,17]. This information in combination with the ability to diffuse with the utilization of aquaporin proteins, makes H_2_O_2_ a viable signaling molecule causing cascades throughout the cell via oxidation of proteins [18]. In normal tissue, H_2_O_2_ is used for physiological responses such as metabolism, cell proliferation, and immune cell recruitment [19]. Although necessary for normal physiological functions, when the concentration of H_2_O_2_ is too high (>100 nM), it can disrupt redox signaling, causing pathophysiology [19]. In cancer, this increased amount of H_2_O_2_ can lead to cancer metabolism and metastasis [20,21].

Mitochondria are arguably the most important organelle within a cell. They are known for their involvement in making ATP using the tricarboxylic acid cycle (TCA) and the electron transport chain (ETC). These organelles are implicated in many other important processes such as cell death, immunological responses, and radiation-induced signaling [22]. Although necessary for cellular function, mitochondria also have implications in many pathologies [23]. Under physiological conditions, ROS are mostly produced at complexes I and III of the ETC [24,25]. These ROS can further damage the mitochondria by causing protein and lipid oxidation within them, as well as outside of the mitochondria [9,26]. Cells can increase their mitochondrial mass using fission/fusion processes during mitochondrial biogenesis [27]. Although this is thought of as the main pathway of increasing mitochondria, there is a novel process called mitochondrial transfer in which mitochondrial particles or pieces are transferred from cell to cell. This can happen in a variety of ways such as tunneling nanotubing, dendritic transfer, naked mitochondrial particles, and extracellular vesicles [28,29].

Extracellular vesicles (EV) are membrane bound particles that can carry cargo from cell to cell. These particles come in three main forms, exosomes, microvesicles, and apoptotic bodies. These subtypes differ in their size, route of generation, and markers on the membrane. Exosomes are generated by the intake of the limited multivesicular body, which is then released into the extracellular environment. Microvesicles are formed by the budding of the cellular plasma membrane. Lastly, apoptotic bodies are formed when the cell is undergoing apoptosis and releasing their contents [30,31,32,33]. Extracellular vesicles can carry a wide variety of cargo, including RNA, mitochondria and mitochondrial contents [34,35,36,37]. Cargos of EVs have been proposed to use as a non-invasive tool for the detection, characterization, and monitoring of cancer such as prostate cancer. Proteo-transcriptomic analyses of EVs from serum of patients reveal enrichment of RNA-proteins, RNA-RNA complexes, and markers which are exclusively expressed in prostate tissues [38]. Additionally, recent study by Chen Tzu-Yi et al. demonstrated that mitochondrial inner membrane proteins are the EV enriched protein signatures that are not significantly different from their cancer cell of origin, implicating a coordinated mechanism between EVs and their donor cells [39].

Cancer reprogramming is an adaptive response mechanism(s) that contributes to cancer survival and progression; including rewired cellular redox state, up-regulated metabolism, as well as activated mitochondrial biogenesis. Our laboratory demonstrated that carboplatin resistant ovarian cancer cells have higher mitochondria function and H_2_O_2_ production levels than the parental ovarian cells [40] while aggressive PCa tissues exhibited redox imbalances via ROS-mediated post-translational modifications of antioxidant enzymes [41]. Previous studies have shown that ROS are generated during RT [42]. Due to the fact that radiation can cause the production of H_2_O_2_, we generated our radioresistant cell line, Clone 695. We measured the amount of H_2_O_2_ produced in Clone 695 cells that survived radiation treatment (both in vivo and in vitro), compared to the parental PCa cell line (PC3 cells). As anticipated, mitochondrial parameters such as mitochondrial respiration, mitochondrial mass, and mitochondrial membrane potential are increased in radioresistant PCa cells. Significantly, we observed EV production upon RT, including EVs carrying mitochondrial contents. In this study, we sought to (1) determine if EVs containing mitochondria is a novel mechanism that contributes to cancer reprogramming via elevated mitochondrial function in PCa cells that survive RT and (2) if H_2_O_2_ could be an arbitrator upon radiation-mediated EV production.

## 2. Materials and Methods

### 2.1. Cell Culture

PC3 cells and PC3 cells that survived radiation treatment (RR-PCa called “Clone 695”) were grown in 5% CO_2_ conditions at 37 C in RPMI-1640 supplemented with 1% glutamine, 10% FBS, Hepes, 1% sodium Pyruvate, and 1% penicillin/streptomycin antibiotics. Tissue culture reagents were purchased from Sigma and Gibco. Cells were passaged less 10 times. PC3 cells (cat.CRL-1435, ATCC, Manassas, VA, USA) were purchased from ATCC. Cell authentication of Clone 695 cells was completed by ATCC using the Cell line Authentication Sample Collection Kit and Service (cat. 70049374) (Appendix A).

### 2.2. Radioresistant Prostate Cancer Cell (Clone 695) Development

PC3 cells were irradiated at 2 Gy/day, 5 times per week, and allowed to rest on the weekends. This process was performed until the total dosage was 66 Gy. Cells that survived 66 Gy radiation were then subcutaneously injected into athymic nude mice. Animal experimental procedures were approved by the Institutional Animal Care and Use Committee of the University of Kentucky (Lexington, KY, USA), approval Protocol No. 01077M2006. Once tumors reached 200–250 mm^3^ in size measured by a caliper (3 times a week), they were irradiated at 2 Gy daily until the total dosage was 10 Gy, which is the dose that suppresses tumor growth established previously in the lab [43]. TrueBeam Varian Linear accelerator (6 MeV) was used for animal radiation (Varian Medical System Inc., Palo Alto, Santa Clara, CA, USA). After about two months, tumors regrew in 2 of the 10 mice (as shown in Figure 1A). These tumors were collected and isolated into a monolayer using a tumor digestion kit from Cell Biolabs INC (cat.CBA-155). Once monolayer cells were established, we performed a colony survival assay to confirm clone 695′s ability to survive radiation.

### 2.3. Cell Growth and Doubling Time Point

Cells were plated at 20,000 cells/well in a corning 12-well clear plate. Cells were collected using trypsin and counted at 24, 48, 72, and 96 h using a hemocytometer in conjunction with trypan blue and bright field microscope (10× magnification). The doubling time point was calculated as follows using the “cell calculator” tool (doubling-time.com; Roth v. 2006):$$\mathrm{Doubling}\;\mathrm{Time}=\frac{\mathrm{duration}\times \log(2)}{\log\left(\mathrm{Final}\;\mathrm{Concentration}\right)-\log(\mathrm{Initial}\;\mathrm{Concentration})}$$

### 2.4. Colony Survival Assay

Cells were plated in a corning 6-well clear plate at 100, 200, 300, and 500 cells per well for 0, 2, 4, and 6 Gy treatment, respectively. Cells were stained with crystal violet and 2.5% paraformaldehyde solution 10–13 days after treatment. Number of colonies was determined by counting colonies of 50 cells or greater. The Plating efficiency (PE) and surviving fraction (SF) were then calculated as follows [44]:$$\mathrm{PE}=\frac{\mathrm{no}.\;\mathrm{of}\;\mathrm{colonies}\;\mathrm{formed}}{\mathrm{no}.\;\mathrm{of}\;\mathrm{cells}\;\mathrm{seeded}}\times \;100\%$$
$$\mathrm{SF}=\frac{\mathrm{no}.\;\mathrm{of}\;\mathrm{colonies}\;\mathrm{formed}\;\mathrm{after}\;\mathrm{treatment}}{\mathrm{no}.\;\mathrm{of}\;\mathrm{cells}\;\mathrm{seeded}\;\mathrm{XPE}}$$

### 2.5. EV Isolation

PC3 cells were plated at a concentration of 5 × 10^6^ cells in a 15 cm corning dish and were irradiated with 0, 6, 46, or 66 Gy. Each dosage represented the following models: control (0 Gy), effect producing dosage (6 Gy), a mid-point to radiation resistant (46 Gy), and radiation resistant (66 Gy) models. The media was changed to media supplemented with 10% exosome free media from System Biosciences (Palo Alto, Santa Clara, CA, USA), 24 h prior to RT. Prior to EV isolation, media from cells was collected 72 h post treatment and filtered using 0.8 µM filter from lifesciences to remove cells and debris larger than 800 nm. Media collected was stored at −80 ℃. EVs were then isolated using the affinity-based method from the Qiagen Exoeasy kit (Germantown, MD, USA). The media collected was thawed and mixed with binding buffer at a 1:1 ratio. This mixture was added to the spin columns and spun at 500 g for 1 min. This process was repeated until all sample/buffer mixture was spun to allow EVs to bind to the column. Next, 10 mL of washing buffer was added to the columns and centrifuged at max (3095× *g*) for 5 min to rinse off any impurities. Lastly, the spin column was added to a fresh conical tube and 1 mL of elution buffer was added to the columns and spun at 500 g for 5 min. The eluate was then placed back onto the column and spun at max (3095 g) for 5 min. To concentrate isolated EVs, the samples were added to an Amicon ultra 3000 filter, which allowed molecules lower than 3 kDa to pass through (cat # UFC500396) in a volume of ~450 µL and centrifuged for 50 min at 14,000× *g*. To collect the concentrated EVs, the 3000 filter was placed upside-down into a new centrifugal tube and spun at 14,000× *g* for 5 min.

### 2.6. EV Measurements

Concentration and size measurements were performed using the ZetaView Nanotracking Analyzer (Particle Metrics, Munich, Germany). Standard bead size of 110 nm was used for calibration prior to running the samples. Samples were diluted in ultrapure 1× PBS pH 7.4 (EV grade, Thermofisher, Waltham, MA, USA) that was filtered through a 0.2 µm PES filter at a 1:10,000 ratio (0.5 µL sample into 5 mL of EV grade PBS). Diluted samples were loaded at a volume of 1 mL. Rinsing was performed in between replicates and each sample with filtered 1× PBS.

### 2.7. Protein Expression by Capillary Based-Automated Western BLOT JESS

Isolated EVs were lysed by adding a lysis buffer containing 11 µL protease inhibitor, 11 µL EDTA, and 200 µL RIPA buffer at a 1:4 ratio. After adding EV lysis buffer, samples were sonicated for 30 s and placed on ice for one minute for three cycles. After the sonication process, samples were allowed to rest on ice for 30 min. Protein concentrations were obtained using the rapid gold bicinchoninic acid assay (Thermofiser). Samples were then diluted with 0.1× sample buffer to yield 1 ug/µL according to the Jess protocol provided by Protein Simple (San Jose, CA, USA). Briefly, samples were added with 5× fluorescent master mix, boiled, and loaded to the instrument at a 1 µg/µL concentration. The following primary antibodies were used: Peroxiredoxin 3 (Prx3) (Proteintech, Rosemont, IL, USA), Transcription factor A, mitochondrial (TFAM) (Santa Cruz Biotechology, Dallas, TX, USA), NADH Dehydrogenase 4 (ND4) (Abcam, Boston, MA, USA), Succinate dehydrogenase A (SDHA) (Proteintech), mitochondrial cytochrome bc1 Complex III (Proteintech), Thioredoxin Reductase 2 (TrxR2) (Proteintech), Nuclear factir erythriod 2-related factor 2 (Nrf2) (Abcam, Cambridge, UK) Manganese superoxide dismutase (MnSOD) (Upstate, Mt Upton, NY, USA), Catalase (Proteintech), Glutathione peroxidase 4 (GPx4) (Proteintech), and Flotilin-1 (Flot-1) (Novus, Contenial, CO, Centennia, CO, USA). All primary antibodies were used at a 1:20 dilution, unless specific. A chemiluminescent secondary detection system was used with either rabbit or mouse secondary antibodies (Protein Simple, San Jose, CA, USA).

### 2.8. Metabolic Parameters

For the mitochondrial stress test and glycolytic stress test, PC3 cells and Clone 695 cells were plated at 20,000 cells/well in a 96-well plate and either the Oxygen Consumption Rate (OCR) or the Extracellular Acidification Rate (ECAR) were obtained using the seahorse XFe96 analyzer from Agilent. The mitochondrial stress test was performed by measuring the initial OCR followed by 1 µM of oligomycin to inhibit complex V (ATP Synthase) of oxidative phosphorylation. 1 µM of FCCP uncoupler was added to collapse the proton gradient followed by 1 µM of both rotenone and antimycin A to inhibit complexes I and III respectively. Spare Respiratory capacity was calculated from the subtraction of the basal OCR (prior to oligomycin treatment) from the maximal OCR (obtained after FCCP treatment). Lastly, ATP-linked OCR was obtained by subtracting the OCR subsequent to oligomycin treatment from the initial (basal) OCR. For the glycolytic stress test, 10 mM glucose is added to start the glycolysis process. Next, 1 µM of oligomycin was added to inhibit complex V. Lastly, 50 mM of 2-deoxyglucose was added to end the glycolytic process by competitively binding to glucose hexokinase. Glycolysis is measured by the ECAR post glucose-injection. The glycolytic capacity is determined by the ECAR after the addition of oligomycin [40].

To determine actual ATP production, an ADP/ATP ratio assay kit from Sigma was used (cat.MAK135-1KT). Cells were plated into a clear 96-well plate from corning at a concentration of 5000 cells/well. Luminescent readings were taken using the SpectraMax Plus 384 from Molecular Devices.

To measure lactate production, 5000 cells/well were plated into a 96-well clear plate from corning. The media was taken from these cells in a volume of 50 µL and placed into a 96-well plate. A lactate assay kit from Sigma Aldrich (St. Louis, MO, USA) (cat#MAK064) was used and absorbance readings were taken at 570 nm using the spectra max plus 384 from molecular devices.

### 2.9. ROS Measurements

Measurement of H_2_O_2_ released into the media was performed using an Amplex Red kit from Thermo Fisher (Invitrogen, Waltham, MA, USA). Polyethylene glycol (PEG) -Catalase (cat.C4963-2MG) (PEG-Cat) as a negative control was added to the cells at a concentration of 500 U/µL, 24 h prior to the measurement. A volume of 50 µL of media was transferred into another clear 96-well plate. Absorbance readings were taken using the SpectraMax plus 384 at 560 nm 2 h after Amplex red reagent was added. Mitochondrial H_2_O_2_ production was measured using the fluorescent probe, MitoPY1 obtained from Biotechne (Minneapolis, MN, USA). MitoPY1 (50 µM) was added to the cells, protected from light, and incubated for 30 min. Cells were washed with PBS (3 times) and MitoPY1 media was replaced with phenol red free-RPMI media. Fluorescent intensity was measured at 495/515 nm using the spectra max plus 384 from molecular devices.

### 2.10. H_2_O_2_ Treatment

Cells were seeded (~2 × 10^6^ cells/plate) in a 75 cm^2^ flask for 24 h. H_2_O_2_ was added in concentrations of 60, 120 and 240 µM for 24 h. Cells treatment with 500 Unit PEG-Cat were used as negative control. Cells and media were collected for further experiments including NTA analysis and Jess Automated Western Blot.

### 2.11. Mitochondrial Mass Measurements and IMARIS Software

PC3 cells were stained with Mitotracker Green FM (mitochondria, ThermoFisher, Waltham, MA, USA, Cat#M17514), cell mask red (cell membrane, Thermo Fisher Cat#A57245), and hoechst (nucleus, Thermo Fisher Cat#H3570). Images were then taken with the Nikon A1 confocal microscope using a Z-stack (Z range:10.00 µM, Step:1.00 µM). The images from the NIS-Element (v5; Nikon Instruments Inc. Melville, NY, USA) were converted into Imaris files (IMS format) by Imaris rendering software (v9.6; Oxford Instruments, Abingdon, UK) for 3D visualization and DATA processing. Border of each cell was traced based on cell mark fluorescence intensity. Green fluorescent intensity represents mitochondrial mass in each cell was then segmented and quantified based on automated relative fluorescence algorithm provided by Imaris software. The following parameters were collected: Green intensity mean per mitochondrion, volume per mitochondrion based on GFP, Sphericity of mitochondrion, and Ellipticity of mitochondrion.

### 2.12. Electron Microscopy

EV imaging

50 µg of EVs placed into PBS were mixed 1:1 with a 2% glutaraldehyde in freshly made sodium phosphate for a final concentration of 1% glutaraldehyde and fixed overnight at 4 °C. Fixed exosomes in a volume of 5 µL were placed on parafilm with the shiny side of the copper grid and allowed to absorb for 20 min. The grid was then washed twice with 100 µL of filtered water. Samples were then stained for 10 min in 2% Uranyl acetate (UA) in water. Excess stain was removed by touching the edge of the grid to filter paper and washing once with a drop of filtered water. Excess water was removed by touching the edge of the grid to filter paper and setting the grid shiny side up to dry. Lastly, the grid was observed under transmission electron microscopy at 80 kV [45].

2.Cell Imaging

Cells were grown on cover slip for 24 h following by 6 Gy RT. After 24 h post-RT, 2.5% glutaraldehyde fixative in Sorenson’s phosphate buffer, pH 7.4 was placed onto the samples and fixed for 30 min. The glutaraldehyde was decanted and rinse 3 times, 5 min each, with room temperature Sorenson’s phosphate buffer, pH 7.4. Post-fixing was performed by adding Caulfield’s OsO_4_ with sucrose for 30 min at room temperature. The samples were then rinsed with ddH_2_O twice for 5 min each time. Dehydration was performed by adding tissue to a graded series of ethanol (15%, 30%, 50%, 75%, 95%, and 4 times of 100%) for 2 min each. Samples were then added to 100% propylene oxide twice for 5 min. Next, 100% EMBED 812 was added to blue flat-bottom moulds (Ted Pella Inc, Redding, CA, USA, Cat#10504), and sample (cells on cover slip) was added in the corner of the mould cell side up. Mould was then polymerized for 18 h in a 65 °C oven. Moulds and blocks were removed. Hydrofluoric acid (32%) was used to dissolve cover slip (30 min) [46]. Samples were then sectioned, stained with UA, and observed under a TEM.

### 2.13. Extracellular Vesicle Uptake

PC3 cells at 5000 cells/well were seeded in 24 well-black with glass bottom plate (0.170 ± 0.005 mm) (Cellvis, Mountain Veiw, CA, USA, Catalog #P24-1.5H-N) and stained using hoechst (nucleus) and cell mask red (cell membrane). EVs were stained with exo-glow protein-EV labeling kit overnight and treated onto the cells at a concentration of 30 ug/time point. Images were taken using the Nikon-A1 microscope with a CO_2_ chamber at 7 h and at 24 h after EV treatments.

### 2.14. Mitochondrial Treatment

Mitochondria were isolated from PC3 cells (1 × 10^6^ cells) using mitochondrial isolation kit for cultured cells according to manufacturing protocol (Cat#89874, ThermoFisher). Isolated mitochondria (0 ug or 10 ug) were then added onto a separate set of PC3 cells using the mitoception method [47]. The plate was spun at 500 g for 5 min for accelerated mitochondrial uptake. After 24 h, excess mitochondria were removed. Cells were then treated with 2 Gy radiation daily (administered 3 times, total dosage 6 Gy) and colony survival assays were performed 10 days post-radiation. For fluorescence imaging, the mitochondria were stained with Mitotracker Red CMXRos to visualize the uptake of external isolated mitochondria.

### 2.15. Statistical Analysis

Data analyses and graphical displays were generated using GraphPad Prism or SAS 9.4. Pairwise comparisons of the mean difference(s) between the vehicle group and treatment group or cell lines was employed with an unpaired, two-sample t-test; while comparison of the 6 Gy treatment group normalized to 0 Gy values was performed using a one-sample *t*-test. Comparison means in more than two groups was accomplished using one-way ANOVA. A Turkey post hoc test was subsequently applied. *p*-value at ≤0.05 was assigned to be significant.

For analysis of particle counts as a function of diameter size and treatment, a generalized linear mixed model was utilized accounting for repeated diameter measurements within a sample and as well as random intercept for each sample. A negative binomial distribution was utilized to account for zero particle counts. Furthermore, the cube root of the particle counts (the outcome) was modelled with a third-degree polynomial function of the cube root of the particle diameter, the radiation/treatment groups and their interactions; testing for the significance of the interaction terms was performed to achieve a parsimonious model. The model curve in the final graph represents the predicted values for the outcome at each group and diameter combination. SAS version 9.4 was utilized the for statistical modeling.

## 3. Results

### 3.1. Radioresistant PCa Cell Line Clone 695 Shows a Significant Difference in Morphology, Growth Rate, and Resistance to Radiation When Compared to Parental PC3 Cells

In order to elucidate the mechanism behind cancer survival and regrowth after RT (radiation resistant phenotype), we developed a PCa cell line named “clone 695”. This cell line was created using PC3 cells, an adenocarcinoma derived from bone metastasis PCa. These PC3 cells were irradiated at 2 Gy, 5 times a week until the total dosage was 66 Gy. Cells that survived this treatment regimen were continued to be cultured and maintained as described in Materials and Methods and were named radio-resistant PC3 cells (RRPC3 cells). Subsequently, RRPC3 cells were subcutaneously xenografted into nude mice. When tumor sizes reached around 250 mm^3^, the tumors were irradiated at 2 Gy/day until the total dosage was 10 Gy. Although tumors shrunk initially, after 6–8 weeks, some of the tumors regrew (2 out of 10). These tumors were collected, isolated into a monolayer cell line. We then named this cell line “clone 695” (Figure 1A). As shown in Figure 1B, the morphology of clone 695 cell line showed an elongated and fibroblast-like appearance while PC3 cells showed a shorter and more epithelial-like morphology (Figure 1B). Growth curve analysis during log phase showed that clone 695 cells grow slower than PC3 cells with a doubling time point of 40 h (2 fold higher than PC3 cells) (Figure 1B). To confirm that clone 695 cells were more resistant to radiation than the parental PC3 cells, a colony survival assay was performed. These results showed that clone 695 had a higher surviving fraction than PC3 cells at a 2 Gy dosage (2 fold higher), which is a commonly administered dose in the clinic (Figure 1C). These findings suggest that clone 695 cells show a different phenotype from the parental PC3 cells.

### 3.2. Radioresistant PCa Cell Line Clone 695 Shows an Increase in Mitochondrial Mass in Individual Cells

When radiating PC3 cells into RRPC3 cells, around the 38 Gy treatment mark, granular structures were observed within the cells (Figure 2A). To further investigate this phenomenon, a mitotracker green fluorescent probe was used to stain the mitochondria within the cells. PC3 cells were either irradiated with sham (0 Gy) or radiation. Upon imaging, it was observed that there were the same structures observed in the cells under brightfield microscopy (Figure 2(A1,A2)), and these structures were stained positive with mitotracker green using a confocal microscope. These data suggest that the granulated structure within the PC3 cells during radiation resistant development are mitochondria (Figure 2(A3,A4)). Due to an increase in mitochondria observed during RT, we investigated whether clone 695 cells have a higher amount of mitochondrial mass than PC3 cells. Using mitotracker green, confocal images were taken of both PC3 and Clone 695 cell lines. After obtaining these images, they were converted into IMS format using a rendering software called IMARIS. By using automated algorithm, the software allows automated tracking, detection, segmentation and quantification analysis of mitochondria on a per cell basis (Appendix A). A color-coded scale bar on the rendered images indicates the intensity of mitochondrial mass based on green fluorescent intensity (red = high, blue/purple = low) (Figure 2B). As shown in Figure 2C, it was observed that clone 695 cell line had an increase in segmented mitochondrial mass and volume per cell, when compared to PC3 cell line. Quantification data also suggests more spherical and elliptical (oblate) mitochondrion in clone 695 cells than PC3 cells (Figure 2C). Due to an increase in mitochondrial mass and volume, the redox status of Clone 695 cells was further investigated in comparison to PC3 cells.

### 3.3. Radioresistant PCa Cell Line Clone 695 Shows an Increase in H_2_O_2_ Production

Since RT is known to increase the level of ROS within the cell, the amount of H_2_O_2_ released into the media was measured using an Amplex red assay. As shown in Figure 3A, clone 695 cells had a significantly higher level of H_2_O_2_ released into the media than PC3 cells. Since ROS production correlates with mitochondrial electron transport chain, specifically mitochondrial membrane permeability, the mitochondrial membrane potential was measured using a TMRE fluorescence probe. Clone 695 cells showed a significantly higher mitochondrial membrane potential than PC3 cells (Figure 3B). To determine where this H_2_O_2_ was produced within the cell, a MitoPY1, a fluorescence probe that is specific to mitochondrial H_2_O_2_ was incubated with the cells (Figure 3C). It was found that clone 695 shows a significantly higher level of mitochondrial H_2_O_2_. This suggests that the increase in H_2_O_2_ within the cell is, at least in part, coming from the mitochondria.

### 3.4. Elevation of Mitochondrial Respiratory Activity Correlates with Reprogramming of Mitochondrial Homeostasis in Radioresistant PCa Cell Line Clone 695

With an increase in mitochondrial H_2_O_2_ as well as mitochondrial membrane permeability, we further tested if there are the differences in mitochondrial respiratory activity between clone 695 cells and PC3 cells. In performing a Seahorse mitochondrial stress test, clone 695 cells show a higher basal OCR, proton leak, and ATP-linked OCR but not spare respiratory capacity, when compared to PC3 cells (Figure 3D). Although clone 695 cell line has a higher amount of ATP-linked OCR, however, this value is calculated on the OCR and is not an actual measurement of ATP. To measure actual ATP production, an ATP/ADP ratio kit was used. As shown in Figure 3E, clone 695 cell line has a higher ATP production than PC3 cells. To investigate if an increase in ATP mainly derived from the mitochondria, a glycolytic stress test was performed using Seahorse instrument. As shown in Figure 3F, ECAR levels in clone 695 cells were not significantly different from PC3 cells. In addition, clone 695 cells showed no difference in lactate production when compared to PC3 cells (Figure 3G). These data suggest that the increase in ATP production is derived from the processes within the mitochondria. Overall, these data imply energy metabolism reprogramming associated with an increase in mitochondrial mass and function in radioresistant PCa.

### 3.5. Extracellular Vesicles Are Released upon Radiation Treatment

Due to the significance of EVs as cargo carrier, we next tested if EVs were being produced upon RT. PC3 cells were irradiated with 6 Gy, media was collected at 72 h post-RT, and was subsequently isolated using Exoeasy kit as described in Materials and Methods. As shown in Figure 4A,D, ZetaView nano tracking analysis show that the EV concentration increased (~50%) post-RT, with average EVs size being about ~150–200 nm. We further compared EV concentration and distribution of PC3 cells and clone 695 cells; as expected, clone 695 cells showed an increased in EVs concentration vs. PC3 cells (Figure 4E). As a complementary method to confirm EV release upon RT, TEM was employed (Figure 4B). Ultrastructural morphology photographs demonstrated the membrane bound vesicles of EVs from both RT versus non-RT PC3 cells (Figure 4C). These data confirm that EVs are being released from PC3 cells upon RT.

### 3.6. Extracellular Vesicles Released during Radiation Treatment Carry Mitochondria as Cargo

Since we observed the increase in mitochondrial mass and mitochondrial function in radioresistant clone 695 cells, we tested if EVs released after RT contain mitochondria. Upon EM imaging of PC3 cells subsequent to RT, it was observed that not only are EVs being released, but they also contain membranous cargo. Upon closer inspection, these membranous cargos have mitochondria-like structures (Figure 4D).

### 3.7. Upon Radiation Treatment, PC3 Cell Line Derived Vesicles Contained H_2_O_2_-Responsive Proteins and Mitochondrial Proteins

To further identify mitochondrial components in RT-derived EVs, we measured mitochondrial proteins in EVs. Expression level for each protein was normalized to total protein loading (Appendix A). As shown in Figure 5 upon 6 Gy RT there was an increase in both mitochondrial DNA and nuclear DNA-encoded mitochondrial proteins within RT-derived EVs. Proteins were found in these vesicles including mitochondrial protein transcription factors such as TFAM as well as mitochondrial electron transport proteins; ND4 (complex I), SDHA (complex II), and Cytochrome bc1 complex (complex III) (Figure 5). In addition to mitochondrial proteins, it was also observed that that upon RT, there was an increase of mitochondrial H_2_O_2_-modifying proteins such as GPx4 and Prx3 (Figure 5). Consistently, both mitochondiral proteins and H_2_O_2_-responsive proteins were increased in EVs derived from PC3 cells treated with hyper fractionated RT, 2 Gy × 23 and 2 Gy × 33 (Appendix A). The increase of these mitochondrial proteins, particularly mitochondrial antioxidants in RT-derived EVs, suggest that RT upregulates H_2_O_2_ production.

### 3.8. Radiation Induces Mitochondrial H_2_O_2_ Production in PC3 Cells

Due to an increase in H_2_O_2_ (Figure 3) in radioresistant PCa cells and the H_2_O_2_ responsive proteins in the RT-derived EV (Figure 5), a MitoPY1 stain was performed to determine if there was an increase in mitochondrial H_2_O_2_. As shown in Figure 6, there was an increase in H_2_O_2_ production within the mitochondria beginning around 6 h post RT and peaking at about 48–72 h post RT. These data suggest that mitochondria contributed to the H_2_O_2_ increased during RT of PC3 cells.

### 3.9. H_2_O_2_ Mediates Extracellular Vesicle Release and Impairing Mitochondrial Function

Due to the production of H_2_O_2_ and H_2_O_2_-responsive proteins being released into the EVs upon RT, H_2_O_2_ could act as signaling molecule that mediates EV release. To test the concept, H_2_O_2_ (60 µM, 120 µM and 240 µM) were added to PC3 cells for 24 h and EVs were then isolated from the media. In analyzing the EVs collected from the H_2_O_2_ treated cells, it was observed that there was a significant increase in concentration of EVs released in the H_2_O_2_ treatment groups (Figure 7A). Size distribution demonstrated smaller EVs upon H_2_O_2_ treatment compared to the untreated control, specifically at 120 µM and 240 µM of H_2_O_2_. (Figure 7B). Importantly, H_2_O_2_-derived EVs contain an increase in mitochondrial proteins as cargo (Figure 7C), similar to RT-derived EVs (Figure 6). In performing a seahorse mitostress test, it was shown that PC3 cells had a lower OCR and spare respiratory rate, with no change in proton leak, when treated with 120 µM H_2_O_2_ (Figure 7D). Pre-treated PC3 cells with PEG-CAT rescued H_2_O_2_-mediated mitochondrial function impairment and H_2_O_2_-activated EV production.

### 3.10. Uptake of External Mitochondria Correlates with an Increase Mitochondrial Mass and Cancer Survival Post Radiation

Since an increase in mitochondria in the form of EVs were observed in PC3 cells post-RT, it was hypothesized that cells that survived RT could uptake these vesicles and utilize mitochondria to increase their chances of surviving post-RT. As shown in Figure 8A, incubation of irradiated PC3 cells with GFP labeled EVs (derived from PC3 cells treated with RT) demonstrated an uptake of EVs into the cells at around 7 h as indicated by GFP localized inside the cells (purple membrane). Quantification of EV number demonstrated the uptake of EVs was greater observed after a 24 h incubation compared to 7 and 0 h (control). Interestingly, GFP intensity of EVs are stable between 7 and 24 h suggesting photobleaching or utilization of EVs by the recipient cells. Subsequent to EV uptake, we next tested if taking up external mitochondria would protect PC3 cells under stressed conditions. PC3 cells were pre-treated with 0 ug or 10 ug of mitochondria according to mitoception protocol [47], prior to RT (0 Gy or 2 Gy × 3). External mitochondria were taken up by irradiated PC3 cells as early as 1.5 h as indicated by an increase in red fluorescence intensity (4 fold, Figure 8B). Subsequently, a colony survival assay was performed 10–12 days post-RT. We found that in the group treated with both radiation and mitochondria, there was a significant increase in surviving fraction when compared to the control group (0 ug of mitochondria) (Figure 8C). Overall, these data suggest that EVs can be taken up by irradiated PCa cells and acquisition of external mitochondria could potentially mediate redox reprogramming and mitochondrial homeostasis in the recipient cells for their survival and regrow after RT.

## 4. Discussion

Cancer reprogramming including rewired cellular redox state, up-regulation of metabolism, and activated mitochondrial biogenesis; is an adaptive response mechanism(s) that contributes to cancer survival, cancer progression, and response to cancer therapy. Aggressive cancers have also been shown to have specific attributes, such as an increase in mitochondrial mass, ROS, and changes in mitochondrial metabolism [48,49,50]. We analyzed mRNA levels of mitochondrial transcription factor TFAM, which is responsible for mitochondrial biogenesis, from PCa patients with various stages (*n* = 499) compared to normal samples (*n* = 52) from The Cancer Genome Atlas (TCGA) database and found that TFAM mRNA expression levels significantly correlate with PCa Gleason scores (Gleason scores 8 > 7 > 6) (Appendix A). Further survival analysis based on TFAM mRNA expression level (cut off at 4.71) demonstrates that PCa patients who have lower TFAM expression are likely to be disease free, when compared to PCa patients who have high TFAM expression, at >0.3 year time frame. Overall, these data suggest that mitochondria could potentially play a role in the aggressiveness of prostate cancer and disease free PCa patients.

Mitochondria have been implicated in many cases of aggressive cancer types with an overexpression of mitochondrial cytochrome C oxidase II being involved with poor prognosis of breast cancer and frequent mitochondrial mutations in PCa [51,52] It has also been shown that mitochondria and their DNA directly play a role in the malignant and tumorigenic transformation of PCa [53,54]. Herein, we generated clone 695 cell line to mimic cancer cells that survived and regrew after RT to test if mitochondria play role in the development of cancer resistance to RT. Upon utilization of a confocal microscope, we found that an increase in small granule structures upon RT, were most likely mitochondria (Figure 2). Increase in mitotracker green intensity indicated a higher mitochondrial mass and volume in radiation resistant cells at single cell level. Together, we hypothesize that cells with higher mitochondria are the ones that survive and become radio-resistant clone 695 cell line. In performing an Amplex red assay, TMRE stain, and MitoPY1 stain, we observed there was an increase in not only H_2_O_2_ released into the media, but an increase in mitochondrial membrane potential and more specifically in mitochondrial H_2_O_2_ of clone 695 cells compared to PC3 cells (Figure 3A). Given this data, we hypothesized that there would be an alteration of mitochondrial respiration, which is another hallmark of aggressive cancers [49,51]. In performing a seahorse mitostress test, our hypothesis was confirmed. Clone 695 cells showed a higher basal OCR, proton leak, and ATP production. Please note, the increase in mitochondrial activity could be due to an increase in mitochondrial volume per cell of clone 695 cells, not the mitochondria of clone 695 cells are more metabolically active. In fact, a higher proton leak in clone 695 cells, suggesting the mitochondria within these cells are uncoupled and oxygen consumption is being used for processes other than ATP production. This is also consistent with in the finding that clone 695 cell line has a significantly lower spare respiratory capacity. These results suggest that clone 695 cell line is functioning at maximum and cannot convert to another source of ATP as easily as PC3 cells. Increased non-mitochondrial OCR has been attributed to NADPH oxidases within the cell. Specifically for PCa, NOX 1, NOX 2, and NOX 5 have been implicated in the increase of non-mitochondrial OCR [52,55]. The glycolytic stress test and lactate measurements are not significantly different (Figure 3B) between PC3 cells and clone 695 cells. Hence, PCa does not conform to the typical Warburg effect, which is the preference of glycolysis over other processes such as the TCA cycle in even the presence of oxygen [56]. Together, the elevation of mitochondrial function in clone 695 cells is accompanied by an increase in mitochondrial H_2_O_2_ production, mitochondrial membrane potential, and mitochondrial mass. These data suggest that reprogramming of PCa’s mitochondrial homeostasis could allow the cancer to survive and regrow after RT.

H_2_O_2_ is a ROS that works as a signal transduction molecule throughout the cell. One of the ways that this occurs is by reacting with thiol group or cysteines of neighboring proteins [57]. During RT, the superoxide radical is converted into H_2_O_2_ by proteins such as MnSOD [58]. It has been shown by Matsumito et al. that the amount of H_2_O_2_ produced during RT is around 0.1 to 0.25 µmol/L/Gy [59]. Likewise, the amount of H_2_O_2_ that diffuses into the cell at 120 µM is estimated to be 20–35 pM [60]. This allows us to determine the effects of H_2_O_2_ in RT and its role in the production of EVs. EVs are produced through a process called biogenesis. This occurs by either the fusion of the vesicle with the plasma membrane or a multivesicular endosomes (MVEs) [61]. Several studies demonstrated that EVs play an important role for intercellular communication by eliminating toxic molecules from the cells, exchanging cargo between the cells, delivering cargo to the recipient cells, and activate signaling transduction in the recipient cells. We demonstrated that Clone 695 cells produce more EVs than PC3 cells and since clone 695 cells have a higher level of H_2_O_2_, thus H_2_O_2_ may contribute to EV production. As shown in Figure 7, upon H_2_O_2_ treatment, EV production is increased with increasing dose of H_2_O_2_. This data suggests that EVs release is at least, in part, mediated by H_2_O_2_. We posit that H_2_O_2_ creates oxidative imbalances causing the cell to release the vesicles to remove damaged biomolecules such as proteins or carry cargo for cell survival such as mitochondria. It is established in squamous head and neck cancer that not only the cargo of EVs are altered upon RT but these cargo promote migration of recipient cells through promoting chemotaxis induced motility as well as AKT-dependent migration [62,63]. EVs have also shown to mediate therapy resistance by regulation of DNA repair, apoptosis, and the cell cycle within recipient cells [64]. Importantly, Jang et al. have shown that EVs containing mitochondrial proteins were released in melanoma [65]. Guescini et al. demonstrated the presence of mtDNA within exosomes released by astrocytes and glioblastoma cells [62]. Similarly, we observed the EVs released upon RT also contained mitochondrial proteins and antioxidant proteins, particularly H_2_O_2_-responsive proteins (Figure 5, Appendix A). The increase of these mitochondrial proteins in RT-derived EVs suggest that RT promotes H_2_O_2_ production, which in turn could stimulate an increase in EV release. We propose that upregulation of H_2_O_2_ during RT cause mitochondrial respiratory impairment which subsequently mediates EV production with mitochondria as a cargo to maintain mitochondrial quality control by removing damaged mitochondria or deliver mitochondrial contents to recipient cells [63]. More importantly, alteration of mitochondrial respiratory function could further promote mitochondrial H_2_O_2_ production which can be transported across mitochondrial membrane to endoplasmic reticulum via aquaporin 11, allowing redox signal conduct in cytoplasm [66] such as activation of calmodulin-dependent protein kinase kinase (CaMKK) [67], which is responsible for MVE formation [64,68]. Please note, in addition to mitochondria or mitochondrial proteins as cargo, RT-derived EVs could carry miRNA, DNA, cytokines, and other transcription factors; which are not being discussed since they are beyond the scope of this study. Moreover, van der Pol et al., as well as Erdbürgger and Lannigan, showed that different methods of EV analysis and isolation can determine the observable size of these vesicles [69,70]. Despite the sensitivity of TEM’s region selection, sample preparations, and the setting of NTA instrument (not exceed 1000 nm), smaller size EVs (~200 nm) that contain “mitochondria-like structures” have not yet been observed by TEM. For the purpose of this paper in terms of what the EVs are carrying, as long as they contain mitochondrial contents or “mitochondria-like structures”, they will be our subject of interest.

Despite these findings and many others, the mechanisms of how mitochondria are packaged in the EVs and how EVs carrying mitochondrial contents are being utilized by recipient PCa cells for possible repair and survival is mostly unknown [71]. A probable mechanism of action could involve the antioxidants carried within the EVs, activating their anti-stress defense systems as demonstrated by Kahroba et al. [72]. Moreover, release of mitochondrial ROS in the cytosol has been show to induce the changes of gene expression through activation of calcineurin and the Cn-dependent retrograde signaling pathway [73], release metabolites to induce histone acetylation [74], and regulate mitochondrial metabolism (mitochondrial fission and fusion) through AMPK activation [75,76]. Here, in, we propose that these vesicles can be taken up by endocytosis in other (recipient) cells and the recipient cells then utilize these vesicles containing mitochondria to produce ATP, to provide precursors of cellular metabolism, to elevate mitochondrial metabolism and to rewire cellular redox state, which are characteristics observed in PCa that survives RT.

Overall, our data suggests that that H_2_O_2_ promotes the production of EVs carrying mitochondrial proteins and that functional mitochondria enhance cancer survival after RT. As show in Figure 9, the increase of H_2_O_2_ production induced the release of EVs containing mitochondria from donor cells. These EVs containing mitochondria can then be taken up by recipient cells. Recipient cells have shown to have an increase in mitochondrial ROS, OCR, mitochondrial mass, and antioxidant proteins, which are also the shown in radioresistant PCa. Extracellular vesicles released post-RT and the cargo within them could aid in the survival of recipient cells. In conclusion, this novel cancer reprogramming phenomena, which is activated by H_2_O_2_, could (1) provide a better insight into therapy-induced resistant cancer’s underlying mechanism; more specifically the role of mitochondria and mitochondrial transfer and (2) could serve as a target to enhance radiation treatment and prevent cancer treatment failure.

## Figures and Tables

**Figure 1 antioxidants-11-02119-f001:**
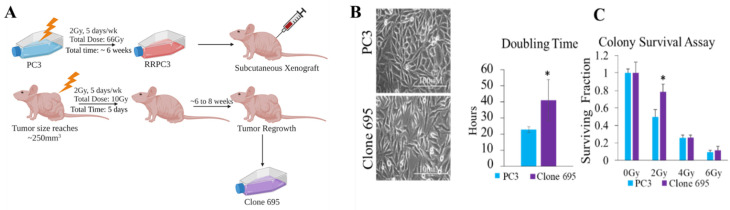
Distinct morphology and survival of Clone 695 cells compared to PC3 cells. (**A**) The process of clone 695 development. (**B**) Cell morphology and cell doubling times of PC3 and Clone 695 cell lines calculated from cell growth curve. (**C**) Colony survival assay of PC3 and Clone 695 cell lines after cells were irradiated at 0 Gy, 2 Gy, 4 Gy, or 6 Gy. * *p*-value < 0.05.

**Figure 2 antioxidants-11-02119-f002:**
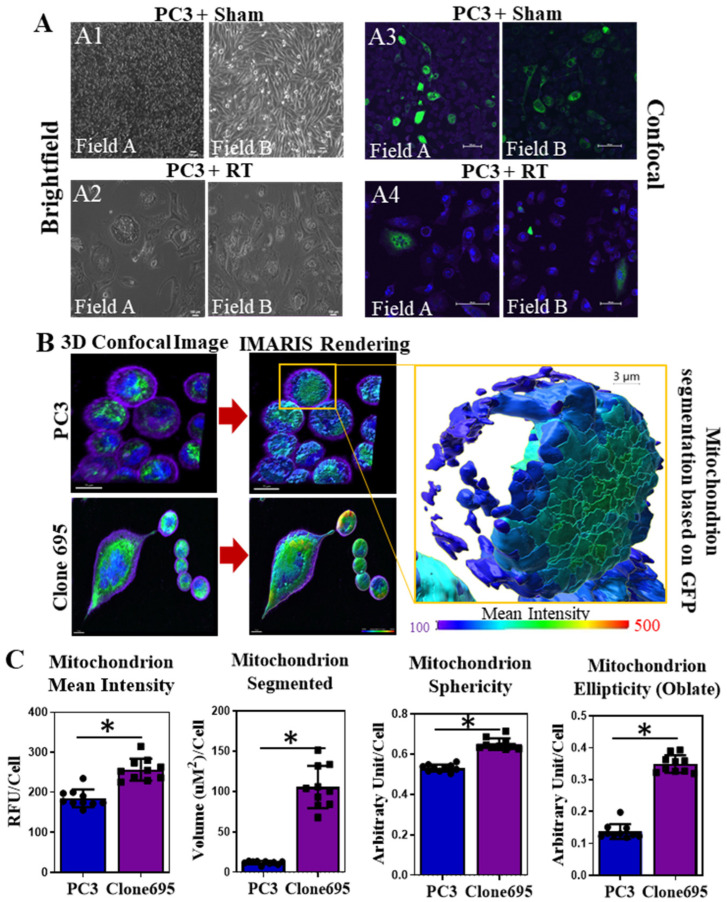
Clone 696 cells show distinct mitochondrial features compared to PC3 cells. (**A**) Cell morphology observed during PC3 cells treated with fractioned RT (clone 695 development). Cells that were exposed to RT altered their morphology with abundance mitochondria observed in cells with larger size. (**A1**) Brightfield images from two different fields and (**A3**) confocal images from two different fields, of PC3 cells taken at 0 Gy RT (sham). (**A2**) Brightfield images from two different fields and (**A4**) confocal images from two different fields, of PC3 cells after treated with 38 Gy RT. Scale bar = 100 µm. Green = Mitochondria, Blue = Nucleus, Purple = membrane. (**B**) Confocal images of PC3 cells and Clone 695 cells converted into IMARIS file for visualization of segmented mitochondria in a single cell. Scale bar = 100 µm. (**C**) Quantitative analysis of each mitochondrion segmented per cell for mitotracker green intensity, mitochondrial volume, mitochondrial sphericity and mitochondrial ellipticity, compared between PC3 cells and clone 695 cells. Circles and squares represent individual data points within PC3 and Clone 695 samples. * *p*-value < 0.01.

**Figure 3 antioxidants-11-02119-f003:**
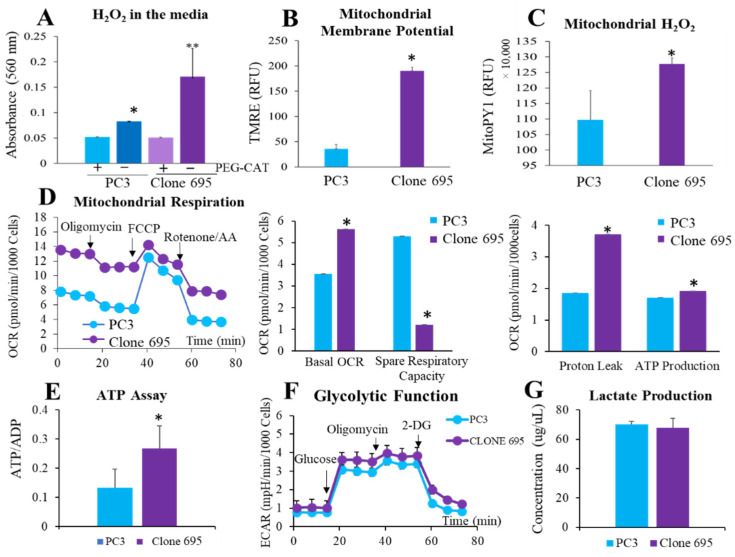
Clone 695 cells shows an increase in H_2_O_2_ production and mitochondrial respiration compared to PC3 cells. (**A**) H_2_O_2_ released into the media after 24 h incubation using an Amplex red assay. PEG-CAT (500 unit) was added for negative control. (**B**) TMRE staining shows mitochondrial membrane potential. (**C**) MitoPY1 intensity indicating mitochondrial H_2_O_2_ production. RFU = Relative fluorescence unit. (**D**) Seahorse Mitochondrial Stress Test. Respiration graphs calculated by response to inhibitors (oligomycin, FCCP, and Rotenone/Antimycin A (AA)). (**E**) ATP/ADP ratio kit showing ATP produced. (**F**) Seahorse Glycolytic stress test. Graph values calculated by response to inhibitors added during assay (Glucose, Oligomycin, and 2-DG). (**G**) Lactate Assay (measuring d-lactate). * *p*-value < 0.05, ** *p*-value < 0.09.

**Figure 4 antioxidants-11-02119-f004:**
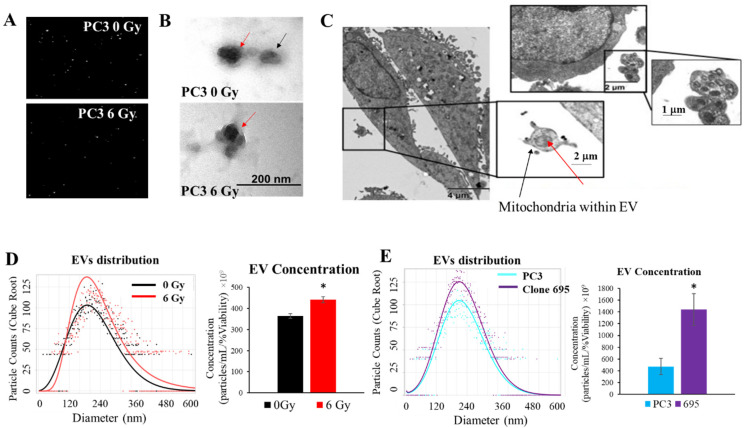
RT activates EVs production. (**A**) NTA images of particles by Zeta View Particle Metrix. (**B**) TEM images of EVs isolated from PC3 cells 0 Gy and 6 Gy. Red arrows denote membrane (outer gray color) and inner cargo (darker gray color). (**C**) EVs carrying “mitochondria-like structures” as cargo upon RT in PC3 cells. Black arrow indicates an Extracellular Vesicle and the red arrow indicates Mitochondria within the vesicles. (**D**) EV count and EV population isolated from PC3 cells at 0 Gy and 6 Gy. (**E**) EV count and distribution isolated from PC3 cells and clone 695 cells, post 72 h after seeding Margin Model from negative binomial overlaid with observed values (Predict values without random effects). * *p*-value < 0.05.

**Figure 5 antioxidants-11-02119-f005:**
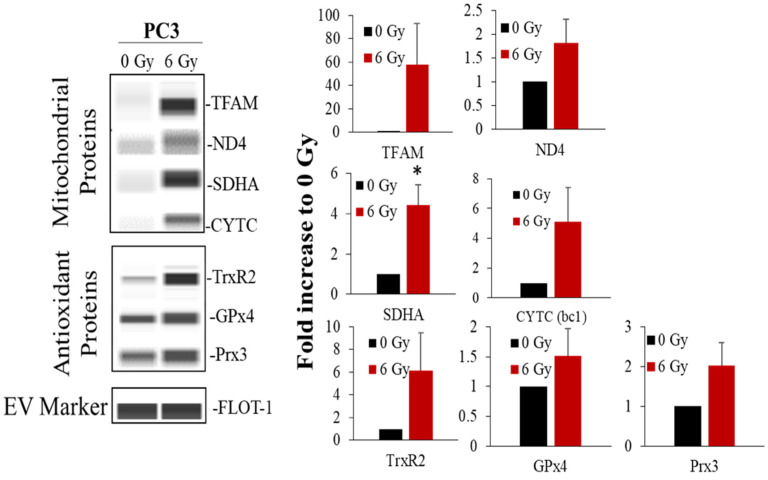
Upon radiation treatment, PC3 cell line derived EVs contained antioxidant and mitochondrial proteins. Mitochondrial OXPHOS proteins and H_2_O_2_-responsive proteins (**left** panel) shown in EVs derived from PC3 cells treated with 6 Gy. Bar graphs indicate average fold changes in protein expression compared to 0 Gy (from at least three separate experiments). * *p*-value < 0.05.

**Figure 6 antioxidants-11-02119-f006:**
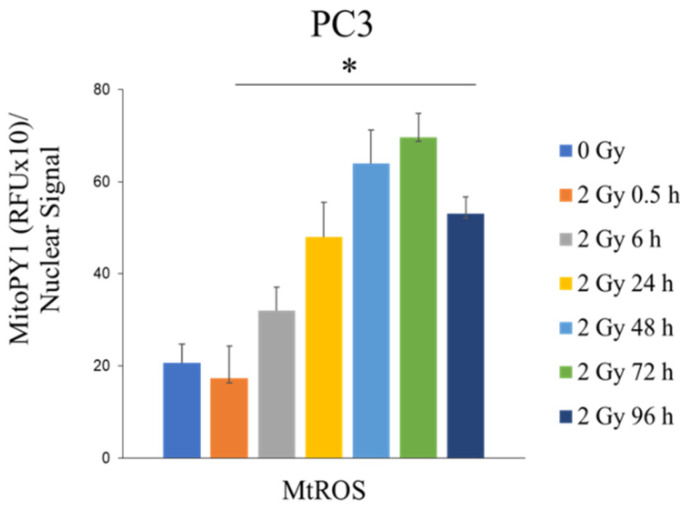
Increase in H_2_O_2_ production upon radiation treatment PC3 cells (6 Gy). H_2_O_2_ production in mitochondria at 0.5, 6, 24, 48, 72, and 96 h post-RT. * *p*-value < 0.05 when compared to 0 Gy.

**Figure 7 antioxidants-11-02119-f007:**
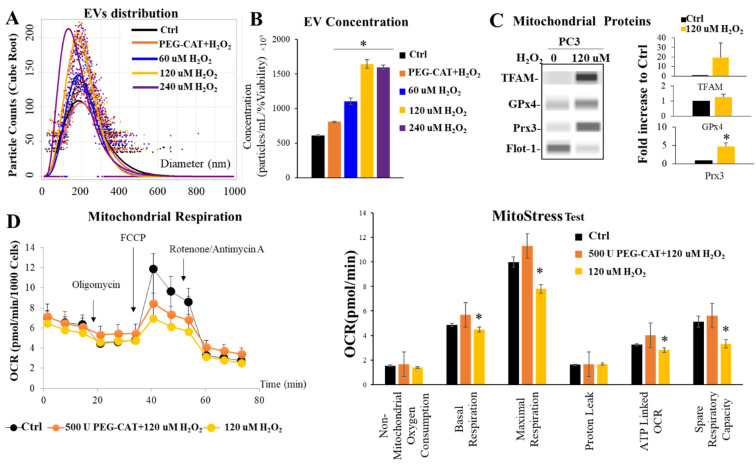
H_2_O_2_- mediated EV release and impaired mitochondrial function. (**A**) Size distribution of EVs derived from PC3 cells treated with H_2_O_2_. Margin Model from negative binomial overlaid with observed values (Predict values without random effects). (**B**) Concentration of EVs that derived from PC3 cells treated with H_2_O_2_. (**C**) Cargo of EVs that derived from PC3 cells treated with H_2_O_2_. Bar graphs indicate average fold changes in protein expression compared to control (Ctrl) (from at least three separate experiments). * *p*-value < 0.05. (**D**) Seahorse mitochondrial stress test of PC3 cell line after treatment of 120 µM of H_2_O_2_. PEG-CAT was used in combination with 120µM H*_2_*O*_2_* as control for all experiments. * *p*-value < 0.05 when compared ctrl and PEG-CAT.

**Figure 8 antioxidants-11-02119-f008:**
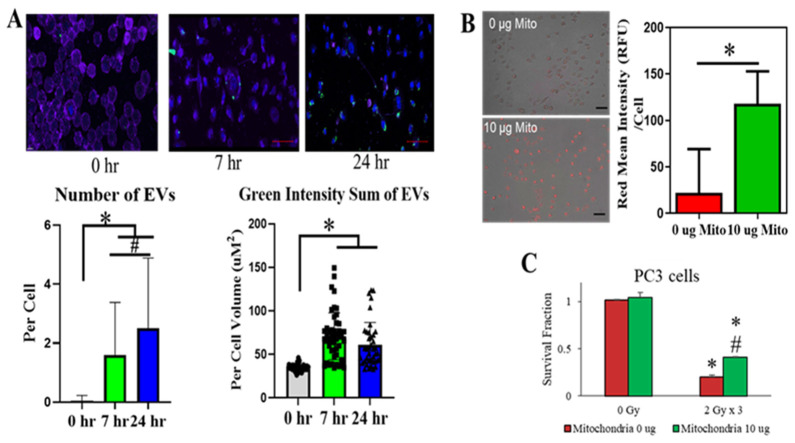
Uptake of mitochondria particles and increase in surviving fraction after RT in PC3 cells. (**A**) EV particles labeled with GFP were uptake by PC3 cells after RT. Green = EV, Purple = plasma membrane, Blue = Nucleus. Number of EVs (based on green GFP) and GFP intensity of EVs were quantify using Imaris software (* *p*-value < 0.0001, # *p*-value = 0.05). (**B**) Isolated external mitochondria (red) were uptake by PC3 after RT. Intensity of RFP represent amount of mitochondria in the cell. Circles, squares, and triangles represent individual data points of 0 h, 7 h, and 24 h respectively. (**C**) Colony survival assay of PC3 cells that uptake external mitochondria. * *p*-value < 0.05 when compared to 0 Gy. # *p*-value < 0.05 when compared to non-mitochondrial treatment.

**Figure 9 antioxidants-11-02119-f009:**
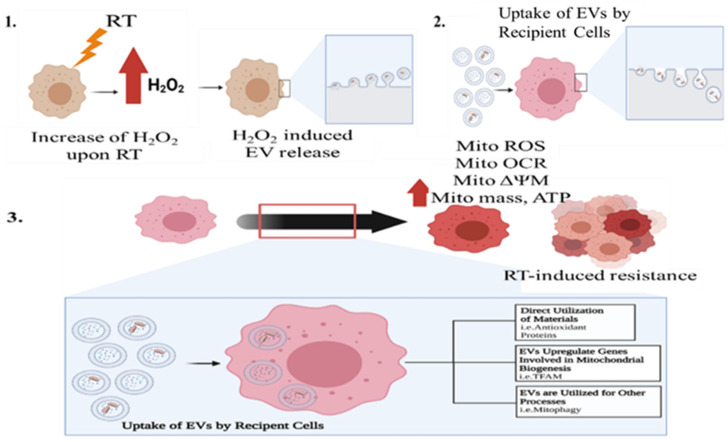
Proposed mechanism of how RT–derived H_2_O_2_ (**1**) Reprogram mitochondrial homeostasis in the recipient cells through EVs containing mitochondria (**2**), which allows the cancer to survive and regrow after RT (**3**).

## Data Availability

Not applicable.

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
