# Peer review of "Hydrogen Peroxide Promotes the Production of Radiation-Derived EVs Containing Mitochondrial Proteins"

_antioxidants, 2022, doi:10.3390/antiox11112119_

Round 1

Reviewer 1 Report

In this manuscript, Miller et al. show that H2O2 promotes the production of EVs carrying mitochondrial proteins. The role of mitochondria and EVs in cancer is studied, both are major advances in the field. This is an important discovery and should be published after suggested revisions. Author’s finding in fig.3 and Fig. 7 are particularly impressive and clearly show the role of H2O2 and mitochondria in PC3 cancer cell lines. However, some of the claims made need to be adjusted (please see below). On average, EV size is shown to be ~200nm. How can an intact mitochondria (~500nm) be encapsulated in EVs? However, authors do show mitochondrial proteins in the EVs, which is a major finding. Below are the suggested changes that need to be addressed and after which, I support the publication of this manuscript. I support the publication of this manuscript, after the suggested changes. No new experiments are needed

Line 89: Extracellular vesicles (EV) are double membrane bound particles. Cite

Double membrane is debatable because EVs carry bo.th

Line 95: Extracellular vesicles can carry a wide variety of cargo, Cite

The authors should refer to Recent papers in 2022 that showed

1: distinct proteo-transcriptomic signatures in EVs from cancer cell of origin (including mitochondrial proteins and RNA).

2: EVs in prostate cancer

Line 157: “representedthe” should be represented the

Line 157: “diseases” should disease

Line 317: started with “We analyzed mRNA levels of TFAM”. Authors have not discussed TFAM in abstract or introduction. A one liner about what TFAM is, will help readers understand the reason why authors are looking at the TFAM expression on TCGA.

Line 315:

Sup fig 2 left shows lower expression of TFAM in tumors. This is not explained and is opposite to what authors are claiming in the text. Furthermore, TCGA does not show a strong correlation of TFAM between normal and tumors (Wilcoxon P= 0.8) and with gleason score (P=0.03).

Looking at these data, I would suggest showing TCGA dataset in supplementary or later figures. Because the next figures seem to more impactful with major discoveries. This change is not required for publication but rather a suggestion to enhance the impact of later major findings.

Line 323:  data suggests that mitochondria, or more specifically TFAM, could play a role in the 324 aggressiveness of prostate cancer and disease free PCa patients.

Claim should. Be toned down.

Fig 2A, Scale bars are impossible to read. Fig 2A legends are needed (what are blue and green colors? ). Fig 2A confocal shows PC3+RT has lower green intensity but 2C shows otherwise. This may just be the region of microscopy but make sure that these align with 2C.

Line 387: there is an unwanted “D” in the line.

Line 436: authors claim that “double membrane bound  vesicles of EVs from both RT versus non-RT PC3 cells (Figure 4C).” The EM pictures do not clearly show double membrane. I would suggest to say “membrane bound” or “cup shaped morphology” observed.

451 “Upon closer inspection, these membranous cargos have mitochodria-liked structures (Figure 4D).” Please clearly show the scale bar in the inset.

Line 595:

“Similarly, we observed the EVs released upon RT also contained 595 mitochondrial proteins and antioxidant proteins, particularly H2O2-responsive proteins 596 (Figure 5A, Supplementary Figure 4). “

Authors mention that intact mitochondria is present in EVs. To my understanding, the mitochondria is about ~500nm long. But the authors show average size of EVs at ~200nm. How can 200nm EVs carry 500nm mitochondria? Perhaps the fragments of mitochondria? I do agree that mitochondrial proteins are inside EVs. That leads to the title of the manuscript. “EV containing mitochondria” should be change to “EV containing mitochondrial proteins” or anything more appropriate that aligns with the findings.

Authors should mention Recent papers that showed proteo-transcriptome in EVs from cancer cells (including mitochondrial proteins) and Exosomes in prostate cancer.

Sup fig. 2 does not have label “Supplementary

Sup fig 4, A and b are not labeled on the figure.

Flotillin should be moved to the main figure.

Reviewer 2 Report

The article entitled, “Hydrogen peroxide promotes the production of radiation-derived EVs containing mitochondria” demonstrates that radiation resistant prostate cancer cells have increased levels of hydrogen peroxide that promotes the release of EVs that contain mitochondria, which promote tumor cell survival after radiation exposure. Using a radioresistant cell line it is shown that hydrogen peroxide and mitochondrial content are increased in these cells. However, nothing else was done with these cells, it is not clear why the EV studies were not done with these cells to show that more EVs are made with these cells as compared to standard PC3 cells. There are some other small changes needed in some of the figures. Overall, this is a well conducted study with very interesting findings. Below are specific areas that are needed for improvement of this manuscript:

1.       EV characterization should be done in between clone 695 and parental PC3 cells to further support the hypothesis as the clone 695 cells have more hydrogen peroxide.

2.       Figure 2, should be reanalyzed using mitochondrial amount per cell to normalize (use mitotracker staining) the data as the Clone 695 has more mitochondria than the PC3 cells. It could be that the mitochondria are not as healthy in the 695 clone but because there are more of them it appears that the mitochondria are more metabolically active.

3.       Figure 5 should be quantified, and significance assessed.

4.      Figure 7A and 7B what level of hydrogen peroxide was used in combination with PEG-CAT?

5.       Figure 7C should be quantified, and significance assessed.

6.       Figure 8A needs a baseline control and should be quantified.

7.       Figure 8B should be quantified.

8.       Is it possible to add in the EV containing mitochondria as shown in Figure 4 to PC3 cells and show that they are protected from radiation killing? This would be more convincing than just adding naked mitochondria.

Minor edits:

1.       Page 12 line 417, should be compared not compare.

2.       Page 12 line 422, should be shown not show.

3.       Page 14 line 486, should be mediates not mediated.

Round 2

Reviewer 2 Report

Discussion page 19 change this sentence: The glycolytic stress test and 625 lactate measurements "are" not being significantly different (Figure 3B) between PC3 cells and 626 clone 695 cells. 

Author Response

Thank you for your response in regards to our revised manuscript, titled “Hydrogen Peroxide Promotes the production of radiation-derived EVs containing mitochondrial proteins”. We are appreciative of reviewer no. 2’s suggestion. We added the word “are” and deleted the word “being” from the sentence “The glycolytic stress test and lactate measurements are not significantly different (Figure 3B) between PC3 cells and  clone 695 cells” in the Discussion section, page 19, line 589. We trust that this change made in this revised version 2 manuscript are satisfactory and we look forward to hearing your final

Thank you for your response in regards to our revised manuscript, titled “Hydrogen Peroxide Promotes the production of radiation-derived EVs containing mitochondrial proteins”. We are appreciative of reviewer no. 2’s suggestion. We added the word “are” and deleted the word “being” from the sentence “The glycolytic stress test and lactate measurements are not significantly different (Figure 3B) between PC3 cells and clone 695 cells” in the Discussion section, page 19, line 589. We trust that this change made in this revised version 2 manuscript is satisfactory and we look forward to hearing your final decision.

Sincerely,

Luksaan Chaiswing
